# Giant barocaloric effects over a wide temperature range in superionic conductor AgI

Araceli Aznar [1], Pol Lloveras [1], Michela Romanini [1], María Barrio [1], Josep-Lluís Tamarit [1], Claudio Cazorla [2], Daniel Errandonea [3], Neil D. Mathur [4], Antoni Planes[5], Xavier Moya [4] & Lluís Mañosa [5]

Current interest in barocaloric effects has been stimulated by the discovery that these pressure-driven thermal changes can be giant near ferroic phase transitions in materials that display magnetic or electrical order. Here we demonstrate giant inverse barocaloric effects in the solid electrolyte AgI, near its superionic phase transition at ~420 K. Over a wide range of temperatures, hydrostatic pressure changes of 2.5 kbar yield large and reversible barocaloric effects, resulting in large values of refrigerant capacity. Moreover, the peak values of isothermal entropy change (60 J K$^{-1}$ kg$^{-1}$ or 0.34 J K$^{-1}$ cm$^{-3}$) and adiabatic temperature changes (18 K), which we identify for a starting temperature of 390 K, exceed all values previously recorded for barocaloric materials. Our work should therefore inspire the study of barocaloric effects in a wide range of solid electrolytes, as well as the parallel development of cooling devices.

[1] Departament de Física, EEBE, Campus Diagonal-Besòs and Barcelona Research Center in Multiscale Science and Engineering, Universitat Politècnica de Catalunya, Eduard Maristany, 10-14, 08019 Barcelona, Catalonia, Spain. [2] School of Materials Science and Engineering, University of New South Wales Australia, Sydney, NSW 2052, Australia. [3] Departamento de Física Aplicada (ICMUV), Malta Consolider Team, Universitat de València, 46100 Burjassot, Spain. [4] Department of Materials Science, University of Cambridge, Cambridge CB3 0FS, UK. [5] Facultat de Física, Departament de Física de la Matèria Condensada, Universitat de Barcelona, Martí i Franquès, 1, 08028 Barcelona, Catalonia, Spain. Correspondence and requests for materials should be addressed to C.C. (email: cazorla@unsw.edu.au) or to X.M. (email: xm212@cam.ac.uk) or to L.Mño. (email: lluis.manosa@fmc.ub.edu)

Proposals for environmentally friendly solid-state cooling devices have been inspired by the discovery of both giant magnetocaloric (MC) effects in a variety of magnetic materials[1–4], and giant electrocaloric (EC) effects in a variety of ferroelectric materials[4–6], but the need to generate large driving fields is problematic. Large magnetic fields are expensive to generate, while large electric fields can lead to electrical breakdown. By contrast, it is easy to generate the hydrostatic pressures required to drive larger[4, 7] and more energy efficient[8, 9] barocaloric (BC) effects non-destructively. BC materials have therefore sparked interest from both academia and industry, but materials selection remains rather limited.

To date, giant BC effects have only been experimentally demonstrated near room temperature in a polymer[10], a small number of relatively expensive magnetic materials[11–16], a number of fluorites[17–20], a hybrid perovskite[21] and a small number of ferro/ferrielectric materials[22, 23]. Following the recent prediction of giant BC effects in fluoride-based superionic conductors at very high temperatures[24], we demonstrate here giant BC effects nearer to room temperature in a powder of AgI, which is the prototypical solid electrolyte that was shown to display fast ionic conduction over one century ago[25, 26].

Above the superionic transition temperature $T_0 \sim 420$ K, AgI exists as the α polymorph, in which the iodine anions adopt a body-centred-cubic $Im\bar{3}m$ structure[27, 28], while the interstitial silver cations are disordered across a fraction of the tetrahedral interstices to yield a very large ionic conductivity that it is comparable with the conductivity of the molten state[26]. Each time the sample is cooled after heating just through the transition, the β and the γ polymorphs[29] are formed in the same ratio[30] due to a first-order reconstructive phase transition that is accompanied by a large 5% increase in volume[31–34]. The β polymorph has iodine anions in a hexagonal-close-packed $P6_3mc$ structure[27, 29, 35], the γ polymorph has iodine anions in a cubic-close-packed $F\bar{4}3m$ structure[27, 36, 37]. The interstitial silver cations in γ-AgI and β-AgI are relatively ordered, whereas in α-AgI they are disordered across a fraction of the tetrahedral interstices, such that the entropy difference $|\Delta S_0| = 63 \pm 4$ J K$^{-1}$ kg$^{-1}$ for the α ↔ β + γ transition is large[38–40].

Here we show that AgI displays peak isothermal entropy changes of $|\Delta S| \sim 60$ J K$^{-1}$ kg$^{-1}$, corresponding to adiabatic temperature changes of $|\Delta T| \sim 18$ K, due to moderate changes of applied pressure $|\Delta p| \sim 2.5$ kbar (where we assume ambient pressure to be zero such that $|\Delta p| \sim p$). These giant and reversible inverse BC effects may be driven over a wide range of temperatures below the transition, yielding values of refrigerant capacity[1] (RC) that exceed all values previously recorded for BC materials. The large BC effects in AgI are associated with a pressure-driven phase transition between the coexisting β and γ polymorphs at low pressure, and the α polymorph at high pressure. On increasing pressure at starting temperatures that lie below the thermally driven transition, the pressure-driven β + γ → α transition increases sample entropy by melting the interstitial sublattice of silver cations[41, 42]. This melting arises as a consequence of reducing the cation hopping distances, and increasing the number of vacant interstitial sites[42].

## Results

X-ray diffraction confirmed that our as-received AgI powder comprised a mixture of β and γ polymorphs, but it could not resolve their relative proportions. At atmospheric pressure, calorimetry data obtained on heating and cooling confirmed both $T_0 \sim 420$ K and a large thermal hysteresis of ~25 K (Fig. 1a). The large 5% increase in volume[31–34] for the α → β + γ transition implies large values of d$T_0$/d$p$, tending in the $p \to 0$ limit to −14.0

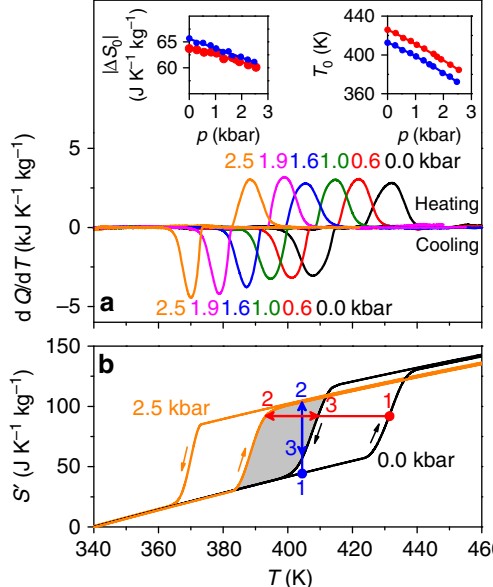

**Fig. 1** Superionic transition in AgI under pressure. **a** Heat flow d$Q$/d$T$ on heating and cooling through the transition at different values of increasing pressure $p$, after baseline subtraction. At pressure $p$, we show (left inset) the thermally driven entropy change $|\Delta S_0|$ and (right inset) transition temperature $T_0$, on heating (red) and cooling (blue) (lines denote fits). **b** Entropy $S'(T,p)$, constructed by plotting $S'(T)$ at $p \approx 0$ and 2.5 kbar, on heating and cooling as indicated via arrows ($S'$ denotes entropy with respect to absolute entropy at 340 K and $p \approx 0$). Adiabatic (isothermal) trajectories in red (blue) are irreversible (1 → 2), or reversible (2 ↔ 3) if wholly within the reversibility region (grey)

± 0.5 K kbar$^{-1}$ on heating and −12.8 ± 0.5 K kbar$^{-1}$ on cooling (right inset, Fig. 1a), as expected[31, 43].

At each measurement pressure, integrating the calorimetry data over temperature yields the entropy change for the thermally driven transition alone $|\Delta S_0| = \left| \int_{T_1}^{T_2} (\mathrm{d}Q/\mathrm{d}T) \mathrm{d}T/T \right|$[7, 22] (left inset, Fig. 1a) (the calculation that yields the temperature and pressure dependence of the absolute entropy (Fig. 1b) is explained later). The value of $|\Delta S_0| = 64 \pm 2$ J K$^{-1}$ kg$^{-1}$ at zero pressure is in good agreement with values previously obtained by experiment[38–40], while the small decrease in $|\Delta S_0(p)|$ with increasing pressure may be quantitatively understood in terms of 'additional' isothermal entropy changes[22] $\Delta S_+(p) = -(\partial V/\partial T)_{p=0} \times p$ away from the transition (this expression for $\Delta S_+(p)$ assumes Maxwell relation $(\partial S/\partial p)_T = -(\partial V/\partial T)_p$). Plots of $V(T)$[33, 34, 44, 45] imply that these 'additional' entropy changes $\Delta S_+(p)$ are negligible at temperatures below the transition, while at temperatures above the transition they are small and conventional such that $\Delta S_+(0 \to p) < 0$.

Given that we identify the same zero-pressure value of $|\Delta S_0|$ from a number of heating and cooling runs at zero pressure, we infer that the ratio of the β and γ polymorphs on cooling through the transition is likely to be constant, as expected[30]. Given also that the pressure dependence of $|\Delta S_0|$ (left inset, Fig. 1a) can be explained purely in terms of the finite additional entropy $\Delta S_+(p)$ at temperatures lying above the transition, we infer that the ratio of polymorphs also remains constant at finite pressure. The nominally constant ratio of β and γ polymorphs at any temperature and pressure (where the β + γ phase exists) implies that it is reasonable to use the quasi-direct method[4, 7] later in order to evaluate BC effects associated with the α ↔ β + γ transition. Even if there were a change in this ratio, it should have little influence on these BC effects, as phonon spectra imply that the two polymorphs are separated in entropy by a small amount (~1%) that varies little with pressure (see Methods and Fig. 2).

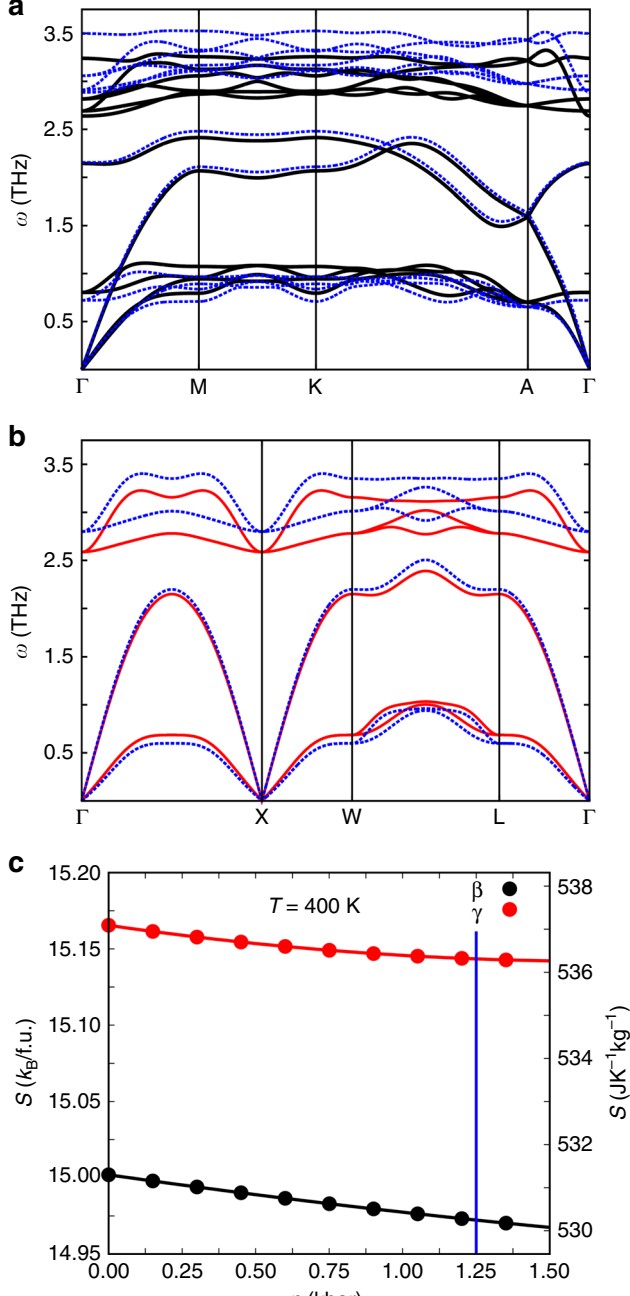

**Fig. 2** First-principles calculations of AgI polymorph entropy. For the **a** β and **b** γ polymorphs in the β + γ phase, we show phonon dispersion spectra at the equilibrium volume (solid black and red), and on reducing the volume by 2% (dashed blue) ($\omega$ denotes the angular frequency of vibration on passing between the labelled points of the Brillouin Zone). **c** Vibrational entropy $S$ for the β (black) and γ (red) polymorphs at 400 K, normalized by formula unit (f.u., left axis) and mass (right axis). Experimentally, a pressure of 1.25 kbar drives the transition at 400 K (vertical blue line)

Sample entropy $S'(T, p)$ was evaluated with respect to the absolute entropy at a given temperature $T_R$ below the transition as,

$$S'(T, p) = S(T, p) - S(T_R, p) = \int_{T_R}^{T} \left( c(T) + \frac{dQ}{dT} \right) dT/T. \quad (1)$$

Given that below the transition $(\partial V/\partial T)_p$ is very small, $S(T_R, p) \simeq S(T_R, 0)$, and taking $T_R = 340$ K, the results for zero and

maximum pressure (2.5 kbar) are shown in Fig. 1b. When integrating over temperatures lying inside the transition region, we used our measured values of $dQ(T, p)/dT$ (Fig. 1a), when integrating over temperatures lying outside the transition region we set $dQ/dT = 0$, and at all temperatures of integration we used the specific heat capacity data $c(T)$ measured[39] at atmospheric pressure ($p \sim 0$). At temperatures lying above the transition region, the decrease of $S'$ with increasing pressure (Fig. 1b) arises as a consequence of the finite additional entropy $\Delta S_+(0 \to p) < 0$ which, as previously mentioned, is due to $(\partial V/\partial T)_p > 0$.

Inverse BC effects driven using our maximum pressure change of 2.5 kbar are only reversible in thermally anhysteretic regions of parameter space (see refs. [46–48]), e.g. in the region of $(S', T)$ space that is bounded by $S'(T, p \sim 0)$ on cooling and $S'(T, 2.5$ kbar) on heating (grey, Fig. 1b). By following adiabatic (isothermal) trajectories denoted red (blue) in Fig. 1b, we see that an irreversible BC effect $(1 \to 2)$, whose trajectory starts outside the reversibility region, is larger than the corresponding reversible BC effect $(2 \leftrightarrow 3)$, whose trajectory lies wholly within the reversibility region.

By likewise obtaining $S'(T, p)$ for our other measurement pressures, we use trajectories such as those described above to identify the maximum values of $\Delta S(T)$ (Fig. 3a, c) and $\Delta T(T)$ (Fig. 3b, d) that may be achieved irreversibly (Fig. 3a, b) and reversibly (Fig. 3c, d) on both applying $(0 \to p)$ and removing $(p \to 0)$ pressure. The maximum values of $|\Delta S| \sim 62$ J K$^{-1}$ kg$^{-1}$ and $|\Delta T| \sim 36$ K that can be achieved irreversibly with $|\Delta p| \sim 2.5$ kbar are reduced to values of $|\Delta S| \sim 60$ J K$^{-1}$ kg$^{-1}$ and $|\Delta T| \sim 18$ K when the constraint of reversibility is imposed. The magnitude of irreversible (Fig. 4a, b) and reversible (Fig. 4e) isothermal entropy changes compares favourably with the values recorded for the best BC materials[11–16, 20–22] whether assuming normalization by mass or volume, especially because some of these literature values contain an irreversible component owing to the fact that hysteresis was not taken into account[11–13, 15, 16, 20]. Separately, our large reversible BC changes in entropy compare favourably with recently theoretically predicted[49] large mechanocaloric changes in entropy in thin films of γ-AgI driven by biaxial stresses up to 10 kbar.

The ability to drive large BC effects over a large ~60 K temperature span (Fig. 3a, b), which arises because of the large shift in $T_0$ with pressure (right inset, Fig. 1a), yields values of refrigerant capacity RC $= |\Delta S_{peak}| \times$ [FWHM of $\Delta S(T)$] that exceed the values reported for all known BC materials[11–16, 20–23], again whether normalizing by mass (Fig. 4c) or volume (Fig. 4d). Here we have used our larger values associated with irreversible BC effects, in order to achieve a fair comparison with the available literature data. However, the value of RC $= 1.1$ kJ kg$^{-1}$ (6.2 J cm$^{-3}$) for $p = 2.5$ kbar computed from our reversible value of $\Delta S(T)$ (Fig. 3c) is also large.

In summary, our observation of giant and reversible inverse BC effects in the prototypical solid electrolyte AgI, near its superionic phase transition, should inspire the wider study of BC effects in similar materials. In future, one may decrease the transition temperature of AgI by chemical substitution[42] or nanostructuring[50]. More generally, it would be attractive to reduce the thermal hysteresis of any given superionic transition in order to increase the magnitude of BC effects, and widen the temperature range of reversibility.

## Methods

**Sample characterization.** Powders of AgI (99.999%) from Sigma-Aldrich were characterized using a commercial TA Q100 differential scanning calorimeter, and a high-resolution X-ray Bruker D8 reflection diffractometer with Cu Kα$_1$ = 1.5406 Å radiation.

**Pressure-dependent calorimetry.** AgI powder mixed with an inert perfluorinated liquid was hermetically encapsulated by Sn. Measurements of heat flow under hydrostatic pressure were performed at approx. ±4 K min$^{-1}$, using a bespoke

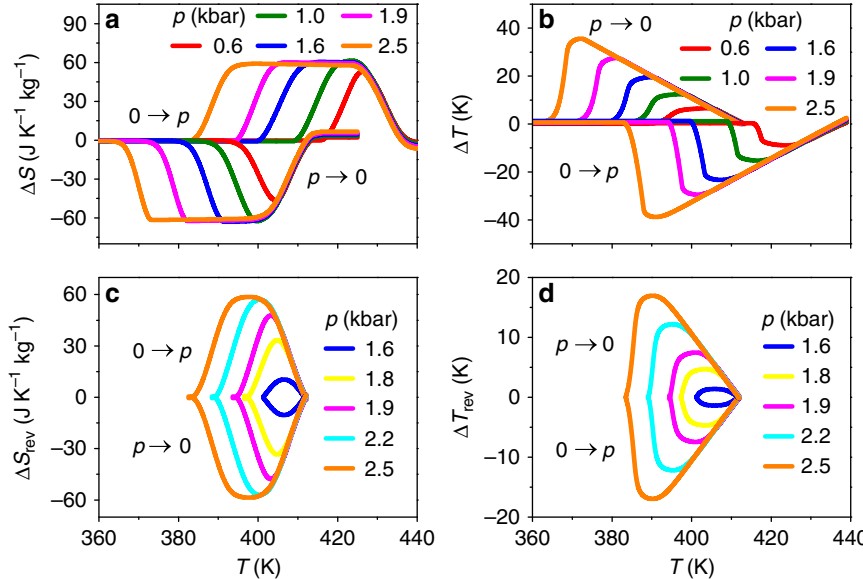

**Fig. 3** Giant inverse BC effects in AgI. **a**, **c** Isothermal entropy change $\Delta S$ and **b**, **d** adiabatic temperature change $\Delta T$ for applying and removing pressure $p$, under **a**, **b** irreversible and **c**, **d** reversible conditions. Data for $p = 2.5$ kbar deduced from Fig. 1b, data for other pressures deduced from analogous constructions

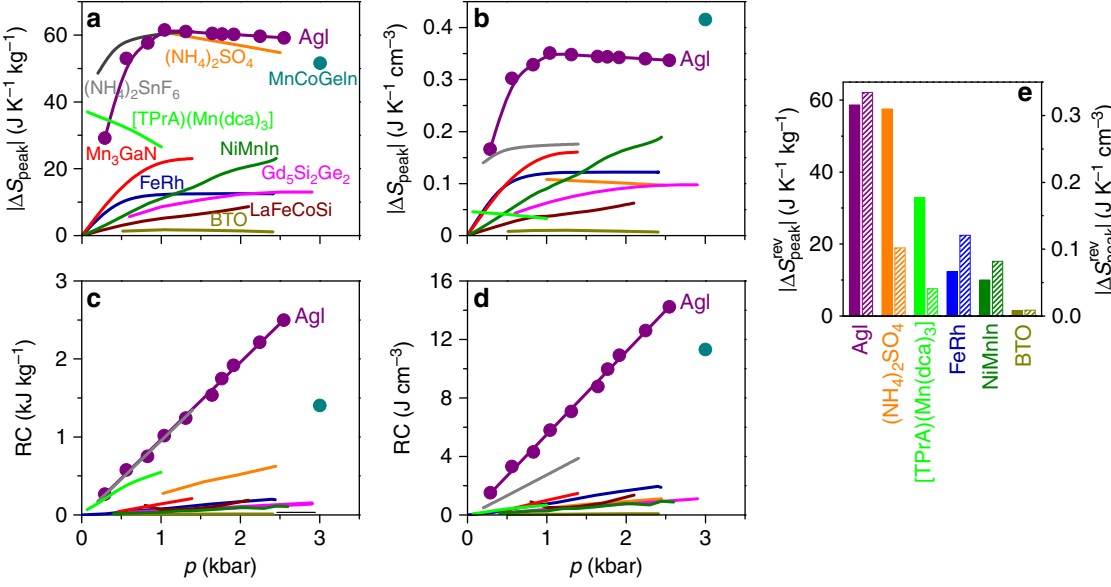

**Fig. 4** Comparison of BC effects. For AgI and other BC materials[11–16, 20–23], we show for pressure changes |$\Delta p$| of magnitude ~$p$ both **a**, **b** peak values of the temperature-dependent isothermal entropy change |$\Delta S_{peak}$|, and **c**, **d** the corresponding refrigerant capacity RC = |$\Delta S_{peak}$| × [FWHM of $\Delta S(T)$], as normalized by **a**, **c** mass and **b**, **d** volume. Solid lines represent fits. **e** The largest values |$\Delta S_{peak}^{rev}$| from **a**, **b** that may be achieved reversibly (solid columns represent mass normalization on left axis, shaded columns represent volume normalization on right axis). For clarity, (NH)$_4$SnF$_6$ alone represents the fluoride compounds[20]. The colour code for BC materials is common to all panels

differential thermal analyser whose resistive heater operates between room temperature and 473 K, and an Irimo Bridgman pressure cell that operates up to 3 kbar with a pressure-transmitting medium of Therm 240 (Lauda).

**Phonon dispersion curves.** First-principles density functional theory calculations were performed using VASP[51, 52] and the so-called direct method[53], where components of the force-constant matrix are obtained in real-space within the small displacement approximation. For our calculations, we used dense k-point grids for integration within the Brillouin Zone, large supercells with 256 atoms to guarantee negligible force-constant components at their boundaries, and we considered both positive and negative atomic displacements in order to obtain null values of acoustic phonons at the Brillouin zone centre[54, 55].

**Entropy of the β and γ polymorphs at 400 K.** We first treat thermal effects within the quasi-harmonic (QHA) approximation[55, 56] by writing the vibrational contribution to the Helmholtz free energy as:

$$F_{vib}(V, T) = k_B T \sum_{q,s} \ln\left[2\sinh\left(\hbar \omega_{qs}/2k_B T\right)\right], \qquad (2)$$

where $\omega_{qs}$ represent the vibrational phonon frequencies of the crystal calculated at fixed volume $V$, and the subscripts q and s run over wave-vectors that span the Brillouin Zone and phonon branches, respectively.

The resulting Helmholtz free energy of the crystal is given by:

$$F(V, T) = E_{static}(V) + F_{vib}(V, T), \qquad (3)$$

where the zero-temperature energy contribution $E_{static}(V)$ is calculated by

considering the atoms fixed in their crystalline lattice sites. Calculating $F$ over a grid of volume and temperature points yields an estimate of pressure $p(V,T) = -\partial F(V,T)/\partial V$ and entropy $S(V,T) = -\partial F(V,T)/\partial T$, which permits the entropy of the crystal to be expressed as a function of pressure and temperature. We assume that this vibrational entropy represents total entropy, as the electronic entropy is negligible in light of the fact that AgI is a non-magnetic wide band-gap semiconductor.

**Data availability**. All relevant data are available from the authors.

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

## Acknowledgements

This work was supported by the MINECO, AEI, FEDER and AGAUR project nos. MAT2016-75823-R, MAT2016-75586-C4-1-P, MAT2015-71070-REDC, FIS2014-54734-P, 2014SGR-581 and the ERC Starting grant no. 680032. C.C. acknowledges financial support from the Australian Research Council's Future Fellowship funding scheme (no. FT140100135). Computational resources and technical assistance were provided by the Australian Government and the Government of Western Australia through Magnus under the National Computational Merit Allocation Scheme and The Pawsey Super-computing Centre. X.M. is grateful for support from the Royal Society.

## Author contributions

L.M., J.L.T., C.C., P.L. and X.M. conceived the study and planned the research. A.A., P.L. M.R. and M.B. performed the experiments and D.E. and C.C. performed the theoretical

calculations. Results were discussed by L.M., J.L.T., C.C., P.L., A.P., N.M. and X.M. The manuscript was written by L.M., N.M. and X.M. with substantial input from all other authors.

## Additional information

**Competing interests:** The authors declare no competing financial interests.

