## [Peer Review File · Nature Communications]

Reviewers' comments:

Reviewer #1 (Remarks to the Author):

Aznar et al discuss about the barocaloric effect at the superionic phase transition of AgI. In this unique investigation, the authors paid attention to a sublattice-melting and they demonstrate that this sub lattice melting is also affected by external pressure, as similar to overall (conventional) melting phenomena (like the ice-water melting) induced not only by temperature but also pressure. In addition, they focused on the sublattice weakening and resultant change in an anharmonicity of special vibration modes and they prove that these changes are essential for entropic change at this transition.

In my opinion, this unique paper contains enough valuable and original impact, so it may be suitable for a publication at Nature Communications.

Reviewer #2 (Remarks to the Author):

In this paper the authors have the merit to evidence ionic conductors as a possibly novel class of "caloric materials" for solid state cooling. The present study is aimed at determining the entropy change and temperature change achieved by isostatic pressure on the well known compound AgI. The authors are able to demonstrate that the phase transition between the low temperature phases (beta and gamma) and the high temperature one (alpha) is accompanied by a rather high latent heat (entropy change) and that, as the transition can be shifted by an applied pressure, it can generate a large barocaloric effect. This paper evidences the potential of ionic conductors as "caloric materials", a potential that has been overlooked so far. Concerning the submitted manuscript I have a few remarks.

1) In the text it is commented that the entropy change in the alpha phase ($\Delta S_{\alpha}(460K,p)$) results to be negative. This is however the result of using a certain known term $\Delta S_{\alpha}(340K,p)$. From the graph of Fig.1b, showing the entropy at two different pressure values, it appears that the two curves starts from almost the same value at 340K (i.e. $\Delta S_{\alpha}(340K,p)$ almost zero). Has this value been measured? Or is it a reasonable assumption? Or is the value used the one derived from the theory (i.e. Fig.2b)? If the third case is the right one, it should be clearly stated.

2) Given the large hysteresis in the phase transition, the authors correctly distinguish between irreversible (i.e. obtained only for the first time) and reversible (i.e. under cyclic transitions) values. The reversible values are deduced from the (irreversible) temperature scan measurement as shown in Fig.1b. This point is poorly discussed here (for lack of space). As this method is not completely standard in the literature, the authors may add references to previous works on the magnetocaloric effect where this issue have been thoroughly discussed (i.e.

Doi:10.1103/PhysRevB.85.014430, DOI: 10.1098/rsta.2015.0308).

3) As this study finds direct applications for cooling devices, the "reversible" values are those really meaningful, i.e. those in Fig.3 c,d. However, when the authors want to compare AgI with other materials they use the "irreversible" ones... I understand the interest in comparing different materials, but doing so, all the values will be plagued by an unknown hysteresis amplitude which makes the plots of very little use in practice. It would be much better to use depurated values instead. Therefore I would suggest to rethink Fig.4. Taking into account that only panel (a) of Fig.4 is actually readable (as it contains the names of the compounds beside the curves), one suggestion could be to just show panel (a). Furthermore the entropy change could be less influenced by the hysteresis than other parameters like RC.

RESONSE TO REVIEWER#2

1) *In the text it is commented that the entropy change in the alpha phase ($\Delta S_{\alpha}(460K,p)$) results to be negative. This is however the result of using a certain known term $\Delta S_{\alpha}(340K,p)$. From the graph of Fig. 1b, showing the entropy at two different pressure values, it appears that the two curves starts from almost the same value at 340K (i.e. $\Delta S_{\alpha}(340K,p)$ almost zero). Has this value been measured? Or is it a reasonable assumption? Or is the value used the one derived from the theory (i.e. Fig.2b)? If the third case is the right one, it should be clearly stated.*

The fact that in the alpha phase entropy decreases by applying pressure ($\Delta S_{\alpha} < 0$) is independent from the value taken as a reference for our entropy plots. Actually, this is due to the fact that $(\partial V/\partial T)_p$ is positive for the alpha phase. In barocaloric (and general caloric) effects only entropy changes are relevant (as opposed to the absolute entropy values) and therefore it is customary to refer all entropy values to a given entropy at a temperature below the phase transition. We have arbitrarily taken this temperature to be 340 K. Indeed, the absolute entropy at 340 K at atmospheric pressure ($S(340K, p=0)$) can be computed from measured specific heat data as: $S(340 K, p = 0) = \int_{\sim 0}^{340} \frac{c}{T} dT$, which gives a value of 519.5 J/kg K. It is also worth noticing that for temperatures below the transition (in the beta + gamma phases) $(\partial V/\partial T)_p$ is very small implying that $S(340K, p=0) \approx S(340 K, p)$, and for this reason the two curves in Fig. 1b start from almost the same value at 340 K (which is our reference entropy point). In the new version of the manuscript we have rephrased the paragraph to make it more understandable.

2) *Given the large hysteresis in the phase transition, the authors correctly distinguish between irreversible (i.e. obtained only for the first time) and reversible (i.e. under cyclic transitions) values. The reversible values are deduced from the (irreversible) temperature scan measurement as shown in Fig. 1b. This point is poorly discussed here (for lack of space). As this method is not completely standard in the literature, the authors may add references to previous works on the magnetocaloric effect where this issue have been thoroughly discussed (i.e. Doi:10.1103/PhysRevB.85.014430, DOI: 10.1098/rsta.2015.0308).*

We have added new references to the revised version of the manuscript.

3) *As this study finds direct applications for cooling devices, the "reversible" values are those really meaningful, i.e. those in Fig.3 c,d. However, when the authors want to compare ΔS_{α} with other materials they use the "irreversible" ones... I understand the interest in comparing different materials, but doing so, all the values will be plagued by an unknown hysteresis amplitude which makes the plots of very little use in practice. It would be much better to use deputed values instead. Therefore I would suggest to rethink Fig.4. Taking into account that only panel (a) of Fig.4 is actually readable (as it contains the names of the compounds beside the curves), one suggestion could be to just show panel (a). Furthermore the entropy change could be less influenced by the hysteresis than other parameters like RC.*

We fully agree that reversible values are those really meaningful. Unfortunately they are only provided for a few compounds while most of the data given in the literature correspond to irreversible values. We have added a new panel in figure 4 (new panel 4(e)) which compares our reversible peak entropy values to reversible data reported for a few giant BC materials. This new panel complements the comparison of irreversible data given in the previous version. In order to make all panels readable, we have also added a sentence which clarifies that a common colour code is used in all panels to indicate the different compounds.

REVIEWERS' COMMENTS:

Reviewer #2 (Remarks to the Author):

Editorial Note: this Reviewer provided no further comments to the Authors.